# Exploration of the close chemical space of tryptophan and tyrosine reveals importance of hydrophobicity in CW-photo-CIDNP performances

Felix Torres[1], Alois Renn[1], Roland Riek[1]

[1]Laboratory of Physical Chemistry, ETH Zurich, Zuerich, 8093, Switzerland

*Correspondence to*: Roland Riek (roland.riek@phys.chem.ethz.ch)

**Abstract.** Sensitivity being one of the main hurdles of Nuclear Magnetic Resonance (NMR) can be gained by polarization techniques including Chemically Induced Dynamic Nuclear Polarization (CIDNP). Kaptein demonstrated that the basic mechanism of the CIDNP polarization arises from spin sorting based on coherent electron-electron-nuclear spin dynamics during the formation and the recombination of a radical pair in a magnetic field. In photo-CIDNP of interest here the radical pair is between a dye and the molecule to be polarized. Here, we explore continuous wave (CW) photo-CIDNP with a set of ten tryptophan and tyrosine analogues, many of them newly identified to be photo-CIDNP-active, and observe not only signal enhancement of two orders of magnitude for [1]H at 600 MHz (corresponding to 10'000 times in measurement time), but also reveal that polarisation enhancement correlates with the hydrophobicity of the molecules. Furthermore, the small chemical library established indicate the existence of many photo-CIDNP active molecules.

## Introduction

Despite decades of development and impressive technological improvements, sensitivity remains the main hurdle of Nuclear Magnetic Resonance (NMR) spectroscopy and imaging (Ardenkjaer-Larsen et al., 2015). Chemical induced dynamic nuclear polarization (CIDNP) enhances the sensitivity of NMR thanks to out-of-Boltzmann nuclear spin polarization. The first anomalous lines related to CIDNP were serendipitously observed in 1967 independently by Bargon et al. (Bargon et al., 1967) and Ward et al. (Ward and Lawler, 1967). The radical pair mechanism was proposed by Kaptein and Oosterhoff (Kaptein and Oosterhoff, 1969) and by Closs (Closs, 1969) two years after and remains the cornerstone of the CIDNP theory ever since. Kaptein demonstrated that the polarization arises from the formation and the recombination of a radical pair in a magnetic field. The radicals can be generated in different ways such as heating, flash photolysis, and photochemical reaction. The last generation mechanism is so-called photo-CIDNP and is the one presented in this work. In photo-CIDNP, light is shined into the sample where a photosensitizer is excited and can undergo intersystem-crossing towards a triplet state. The triplet state dye reacts with a molecule of interest (M) and forms a radical pair after abstraction of an electron from that molecule. The newly formed radical pair is in a triplet state and cannot recombine due to the Pauli principle. The interplay of nuclear spin dependent electron intersystem crossing into a singlet state allowing the electron back-transfer yields to different radical pair

recombination kinetics depending on the nuclear spin state. Therefore, CIDNP can be used to study transient radicals that are too short lived for EPR (Closs and Trifunac, 1969;Morozova et al., 2008;Morozova et al., 2007;Morozova et al., 2005), to study protein structure (Kaptein et al., 1978) and folding (Hore et al., 1997;Mok et al., 2003;Mok and Hore, 2004), or to study electron-transfer mechanism (Morozova et al., 2018;Morozova et al., 2008;Morozova et al., 2005;Morozova et al., 2003). The radical pair mechanism is extensively described in different papers, that we recommend to the reader for a deeper understanding

(Goez, 1995;Morozova and Ivanov, 2019;Okuno and Cavagnero, 2017;Kuhn, 2013). Robert Kaptein's key role in the development of the theory underlying the CIDNP mechanism is crystallized in the Kaptein rules which capture the theory of CIDNP into a simple equation in order to qualitatively analyze the sign of an anomalous CIDNP line (Kaptein, 1971). According to the Kaptein's rules, considering a radical pair composed of molecules a and b the sign of the polarization on a nucleus i belonging to a is predicted by the following equation:

$$\Gamma_{ne} = \mu\ \varepsilon \Delta g A_i \quad (1)$$

$\Gamma_{ne}$ is the net polarization sign of the radical a, $\mu$ and $\varepsilon$ are Boolean values. $\mu$ is positive when the radical is formed from a triplet precursor, and negative otherwise. $\varepsilon$ is positive for recombination products and negative for the radical escaped or the

45 transfer reaction products. $\Delta g$ is the sign of the g-factors difference between the two radicals, i.e. $g_a$-$g_b$, and $A_i$ is the hyperfine coupling constant sign of the considered nucleus i in the radical a which makes the reaction nuclear spin selective. The Kaptein's rules equation predicts sign of polarization and reflects the complex nature of the reaction path that yields to out-of-Boltzmann spin polarization. Therefore, it can be used for a qualitative analysis of the photo-CIDNP polarized products. Extensive studies of the photo-CIDNP effect monitored by ultra-violet absorbing dyes such as FMN (Tsentalovich et al., 2002),

bipyridyl (Tsentalovich et al., 2000) or 3,3',4,4'-tetracarboxybenzophenone (TCBP) (Morozova et al., 2011) using state of the art Time resolved (TR) photo-CIDNP (Hore et al., 1981) applied to a small well-established list of ca. a dozen or so target molecules such as tryptophan (TRP) and tyrosine (TYR) (Table S1) elucidated a great understanding of the photo-CIDNP theory along with the determinants of the polarization of the individual dye molecule pair systems including magnetic field dependency, g-factors, hyperfine couplings and timing. However, less extensively described molecules are also reported in the

literature and draw the start of the endeavour to explore the chemical space of photo-CIDNP performances (Table S1). However, this number of molecule remain small and would benefit from being significantly increased. In contrast to the physical-based approach, it is the focus of our attempt to elaborate on the chemical space of continuous wave (CW)-photo-CIDNP active molecules in order to bring CW-photo-CIDNP towards a versatile and straightforward applicable tool in biomedical and biochemical research. Initially, our recent efforts have been to push the dye absorption towards more

biocompatible wavelength such as near infrared (650-900 nm). In parallel, the performing of photo-CIDNP with readily handled light source is a contemporary goal (Bernarding et al., 2018). These efforts yielded to the discovery of the Atto Thio 12 (AT12) dye which monitored CW-photo-CIDNP experiments with a promising signal-to-noise enhancement (SNE) after laser irradiation at 450 nm (Sobol et al., 2019). Furthermore, the light source was an affordable continuous-wave laser which

could be setup within a few minutes on different Bruker spectrometers, in our case: 200 MHz Avance, 600 MHz Avance III and 700 MHz Avance Neo. On this journey to establish a CW-photo-CIDNP in biomedicine, a strong dependency of the photo-CIDNP SNE on the dye-molecule couple was observed. For instance, tryptophan is poorly polarized in the presence of AT12 but highly polarized in the presence of fluorescein, and tyrosine is highly polarized in the presence of AT12 and less well polarized in the presence of fluorescein (Okuno and Cavagnero, 2016;Sobol et al., 2019). These changing performances were attributed in an initial approximation to the chemical structures of the aromatic ring and the atoms in their close vicinity yielding to different magnetic parameters, g-value and hyperfine coupling (HFC). This assumption was corroborated by the observation of anomalous line sign alternation for an oxidocyclization product of tryptophan while the dye monitoring the reaction is changed from AT12 to fluorescein (Torres et al., 2021). This observation is related to the g-factor difference between the two molecules as we shall see, and is thus an elegant illustration of the Kaptein's rules.

Moreover, in this work we show on the importance of side-chains in the intensity of the anomalous lines in continuous wave (CW) photo-CIDNP experiments. Although the effect of the chemical modifications on the aromatic part of the molecules are well described by theory and confirmed experimentally (Kuprov et al., 2007;Kuprov and Hore, 2004), the effect of the non-aromatic moieties are considered to be conditioning the triplet state dye quenching kinetics, as observed in Time resolved (TR) photo-CIDNP (Saprygina et al., 2014). While the exact mechanism of polarization can be only evaluated by TR-photo-CIDNP, in the context of biomedical NMR application aiming for the highest polarization at low micromolar molecule concentration, CW-photo-CIDNP appears to be the method of choice suggesting the exploration of a CW-photo-CIDNP-based empirical approach indicated. This work reports on this approach by screening 10 compounds with two different dyes.

**Results**

**1.1 On the Kaptein rule of the oxidocyclization product of tryptophan α-hydroxypyrroloindole**

The distinct photo-CIDNP performances of the different dye-molecule couples was previously discussed in the literature(Sobol et al., 2019;Okuno and Cavagnero, 2016). For interest here, the tryptophan presented a higher signal to noise enhancement (SNE) when polarized upon fluorescein (chemical structure see Figure 1) irradiation when compared with the dye AT12 (chemical structure see Figure 1) as shown in Figure 1 and listed in Table 1 whereas tyrosine was better polarized in the presence of AT12 (Figure 1) (Sobol et al., 2019). The differential effect of the dyes to trigger radical pair mechanism has been further studied and yielded to the serendipitous observation of 3α-hydroxypyrroloindole (HOPI, chemical structure, see Figure 1) an oxidocyclization product of tryptophan which is highly polarized after irradiation in the presence of AT12 (Figure 1, Table 1). The study of HOPI revealed surprising features such as a different polarization yields between the *cis* and *trans* diastereoisomers, and the sign alternation of the anomalous intensities depending on the dye used to form the radical pair (Torres et al., 2021). This sign alternation is here now assessed in the light of the Kaptein rules (eq. 1). In the case of the photo-

CIDNP reaction that is performed for all the experiments of this work, μ is positive since the radical pair is formed in a triplet-state. Moreover, the polarized species are the recombination products of the radical pairs and thus the parameter ε is positive. Hence, the variable parameters are the hyperfine coupling when the molecule changes, e.g. from tryptophan to HOPI, or/and the Δg. The Δg can also alter when the dye used is changed. Hence, for the same molecule a sign change of the NMR signal upon switch from a dye to another one is necessarily caused by an alternation of the sign of the Δg in the Kaptein rule equation

(eq. (1)). As shown in Figure 1, the sign switch is observed in the case of HOPI (evidently for all the resonances) when the dye is altered from fluorescein to AT12. This finding can then be used to set the unknown g-factor of HOPI radicals between the g-factors of the dyes and in respect to the known values of tryptophan and tyrosine radicals as shown in Figure 2. Moreover, the observation of the anomalous lines signs in photo-CIDNP experiments monitored with TCBP in previous study, enabled to rank the HOPI compounds with a g-factor between 2.0034 (fluorescein) and 2.0035 (TCBP) (Torres et al., 2021).

This is not only an elegant illustration of the Kaptein's rules, but also a witness of a g-factor evolution upon chemical modification from tryptophan to its oxidocyclization product, HOPI (Figure 1).  Since the g-factor originates from spin-orbit coupling, the shape of the aromatic ring was suspected to be the main factor for an increased g-factor and improved polarizability. This hypothesis is consistent with the results previously obtained for tyrosine which has a comparable aromatic system and is preferentially polarized in the presence of AT12 (Sobol et al., 2019). Therefore, the photo-CIDNP spectrum of

2-3 dihydro-tryptophan (dH-TRP) which has the same aromatic system as HOPI (Figure 1), was recorded for both the dyes, AT12 and fluorescein (Figure 1). As expected, the polarization sign switch could be observed again upon dye change, confirming the idea that similar aromatic systems should/may yield close g-factors (as pinpointed to in Figure 1 and 2). However, the good photo-CIDNP performance of the HOPI compound is not observed for dH-TRP as the polarization enhancement in the presence of AT12 was only 10-fold and 17-fold in the presence of fluorescein (Table 1). This difference

cannot be attributed to a slight difference in the g-factor (towards the g-factor of fluorescein) because the SNE in the presence of fluorescein did not compensate the loss in SNE in the presence of AT12 as it would be expected if the enhancement would solely rely on a g-factor value change (Figure 1 and Table 1).

**Table 1: Signal enhancement of tryptophan and tyrosine derivatives (100 µM) after irradiation in the presence of AT12 or fluorescein (20 µM) at 1 W for 4 seconds. The reported charges are of the diamagnetic molecules. The values for the tyrosine enhancement are taken from ref 20 and for the tryptophan and HOPI enhancement from ref 22. Nevertheless, sample conditions were identical. The log(P) is the logarithm of the partition coefficient P, and P is the ratio of the concentration of the compounds in a mixture of the two immiscible solvents octanol and water (P=[molecule]$_{octanol}$/=[molecule]$_{water}$).**

| molecule | AT12 SNE | Fluorescein SNE | Charge | Log(P) |
|----------|----------|-----------------|--------|--------|
| TRP | 18[a] | 54[a] | 0 | -1.1 |
| TRPA | 45 | 70 | +1 | 1.6 |
| TYR | -38[b] | -20[b] | 0 | -2.3 |
| TYRA | -63 | -23 | +1 | 1.1 |
| HOPI | 90[a] | -70[a] | 0 | -2.1 |
| dH-TRP | 10 | -17 | 0 | -1.5 |
| PEI | 44 | 103 | +1 | 1.9 |
| IPA | 45 | 70 | -1 | 1.8 |
| IAA | 27 | 54 | -1 | 1.4 |

[a] (Torres et al., 2021); [b] (Sobol et al., 2019)

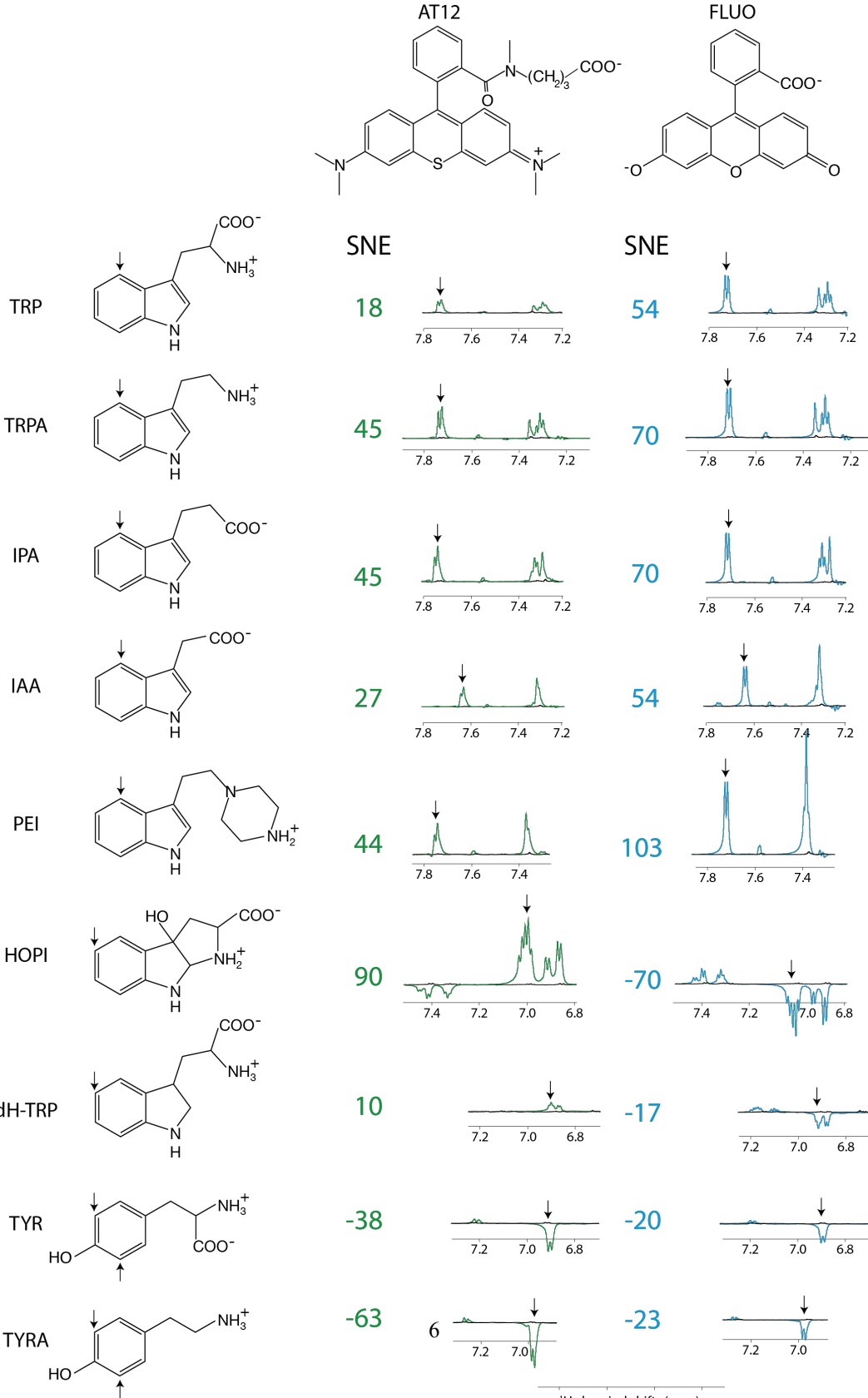

AT12    FLUO

SNE    SNE

TRP    18    54

TRPA    45    70

IPA    45    70

IAA    27    54

PEI    44    103

HOPI    90    -70

dH-TRP    10    -17

TYR    -38    -20

TYRA    -63    6    -23

¹H chemical shifts (ppm)

**Figure 1: Photo-CIDNP spectra of tryptophan or tyrosine derivatives. The best aromatic proton's signal-to-noise enhancement (SNE) is provided and the corresponding signal is pinpointed with an arrow. All spectra were recorded after 4 sec of irradiation at 1 Watt, in the presence of either AT12 (20 μM, irradiation wavelength 532 nm) or fluorescein (20 μM, irradiation wavelength 450 nm), the to-be-polarized molecule is present in concentration of 100 μM. Color code: green are the AT12 monitored SNE values and photo-CIDNP spectra, blue are the fluorescein monitored SNE values and photo-CIDNP spectra, dark are the non-irradiated reference spectra. The structures are provided in their ionization state at experimental pH 7.1.**

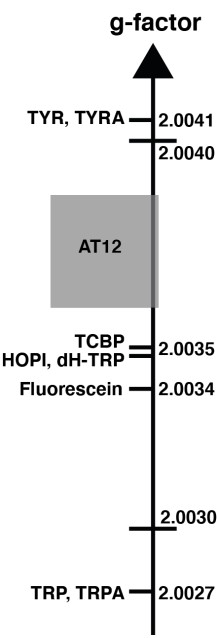

**Figure 2: The sign alternation enables to locate the molecules on the g-values scale. The ranking of HOPI and dH-TRP provides fine information about their radicals g-values. For the non-elsewhere defined abbreviations: TRP stands for tryptophan, TRPA stands for tryptamine, TYR is tyrosine, TYRA is tyramine. (Figure adapted from Sobol et al. (Sobol et al., 2019))**

### 1.2 The involvement of side chain properties in the photo-CIDNP performance of tryptophan and tyrosine derivatives

The lower performances of the dH-TRP in comparison to HOPI's despite similar g-factors turned the focus to the potential involvement of the side chain. Prior work has been done by Saprygina et al. (2014), studying the influence of N-acetylation on the quenching rate of TCBP. The replacement of the α-amine by a N-acetyl, resulted in the vanishing of the positive charge and lower quenching rates of the triplet-state photosensitizer accompanied by lower time-resolved photo-CIDNP enhancements and interpreted as causative of the N-acetylation. Similarly, to this approach, side chains modifications of the same molecular species were studied in the context of CW-photo-CIDNP.

Here, first insights into the potential role of the side chain was gathered by a comparison of the tryptophan-derivative tryptamine (chemical structure see Figure 1) with tryptophan. Tryptamine differs from tryptophan by the absence of the carboxylic acid on its side chain. Indeed, improved CW-photo-CIDNP SNE is observed for tryptamine when compared with tryptophan, especially after irradiation in the presence of AT12 for which a further signal enhancement of a factor of 2 is documented (Figure 1, Table 1). Since both molecules have the same aromatic system and thus similar magnetic parameters (Connor et al., 2008), and a similar reaction mechanism is expected (i.e. ET), the improved polarization of tryptamine might

be due to the charge of the molecule that differs from tryptophan by the absence of the carboxylic acid on its side chain causing potentially a change in the quenching kinetics: Fluorescein's contains a benzocarboxylate moiety of typical pKa 2-2.5 and a xanthenol of pKa 6.4, (Lavis et al., 2007) and is, in the buffer of interest, twice negatively charged. AT12 is neutral in the experimental conditions (pH = 7.1), however the aromatic system is globally carrying a positive charge (Figure 1). Due to its overall positive charge, it is expected that the quenching of fluorescein by tryptamine is faster than by tryptophan, which is globally neutral in the experimental conditions. However, the strongest improvement in terms of CW-photo-CIDNP performances is for AT12 monitored experiments, despite the rather repulsive charges in play.

In order to elaborate further on the hypothesis of the direct potential impact of the charge of the side-chain on the SNE of AT12 monitored CW-photo-CIDNP experiments, such experiments were conducted on tyramine (Table 1), a derivative of tyrosine where the α-carboxylate moiety is absent. Tyrosine and tyramine are preferentially polarized upon irradiation in the presence of AT12 oppositely to tryptophan/tryptamine, due their higher g-factor, as shown in Figure 2. The CW-photo-CIDNP SNE in the presence of AT12 is significantly higher for tyramine when compared with tyrosine (Figure 1, table 1). The minor SNE enhancement for tyramine versus tyrosine photo-CIDNP experiments monitored by fluorescein could be explained by the different Δg. This experiment supports the finding that the chemical modification of side-chains can significantly improve the SNE for CW-photo-CIDNP in the presence of AT12. Next, the CW-photo-CIDNP spectra of indole propionic acid (IPA) and indole acetic acid (IAA) have been recorded. IPA is the negatively charged analogue of tryptophan (Table 1) where the α-amine is lacking. IAA is similar to IPA but the carboxylate group is closer to the aromatic ring, since it is in the β-position. Unexpectedly, the IPA yielded to the same SNE as tryptamine (Table 1) whereas an interpretation of the SNE solely based on the charged, overall negative for IPA, predicted an opposite effect on the performance as compared to the positively charged tryptamine. Despite identical charge as IPA, IAA (Table 1) exhibits comparable performances as compared to tryptophan. Moreover, the 3-(2-(piperazin)ethyl)-indole (PEI) is an analogue of tryptamine where the α-amine is replaced by a piperazin moiety. In PEI, the overall charge is similar to tryptamine since the pKa of the tertiary amine (of the ground state) is close to 4 and only the secondary amine is positively charged, due to its pKa around 9. PEI yielded similar polarization performances as tryptamine upon irradiation in the presence of AT12, and showed even higher SNE for fluorescein monitored photo-CIDNP experiments. An interpretation of these results solely based on the respective overall charges therefore fail to draw any trend. Alternatively, it could be hypothesized that a different side-chains dynamic may play a role in the SNE of CW-photo-CIDNP since with the side chain alterations not only the charge of the side chain changed but also the dynamics with the tryptamine, IPA, IAA, PEI, and tyramine comprise faster side chain motion than tryptophan and tyrosine. This change in dynamics is indicated by the observation that the H$_\beta$ resonances are split for tryptophan and not for the tryptamine, IPA, IAA and PEI (Figure 3). The same degeneracy of the H$_\beta$ chemical shifts is observed when the amine group is removed from the tyrosine to become the tyramine (Figure 3). However, this hypothesis is not supported experimentally since the dihydro-tryptophan which has a higher side-chain mobility as compared to the HOPI shows less polarization than HOPI.

The only summary of this first attempt to interpret the chemical space exploration is that the simultaneous presence of the α-carboxylate and the α-amine is suboptimal for CW-photo-CIDNP SNE when monitored with fluorescein or AT12 as supported by the less good polarization properties of tryptophan, tyrosine, and dH-TRP when compared with their analogues. A corollary of the presence of these α-carboxylate and α-amine is the water solubility of the small molecule and their solvation shell. This idea brings us to another difference between the different side-chains properties, which is hydrophobicity. This can be assessed with the logarithm of the calculated partition coefficient between octanol and water, log(P). The evolution of the hydrophobicity within the different families of compounds and its influence on the CW-photo-CIDNP performances was therefore investigated. Within the tryptophan derivative group, i.e. tryptophan, dihydro-tryptophan, tryptamine, IAA, IPA, PEI, the increasing hydrophobicity is beneficial to the CW-photo-CIDNP performances when monitored by both fluorescein and AT12 dyes (Figure 4A and B). The same trend is suggested for the tyrosine derivatives, tyrosine and tyramine: Tyramine which is more hydrophobic is better polarized, especially in the presence of AT12, than tyrosine (Table 1). The HOPI was not included in this analysis since it is rather far away from the chemical space of the two series of interest.

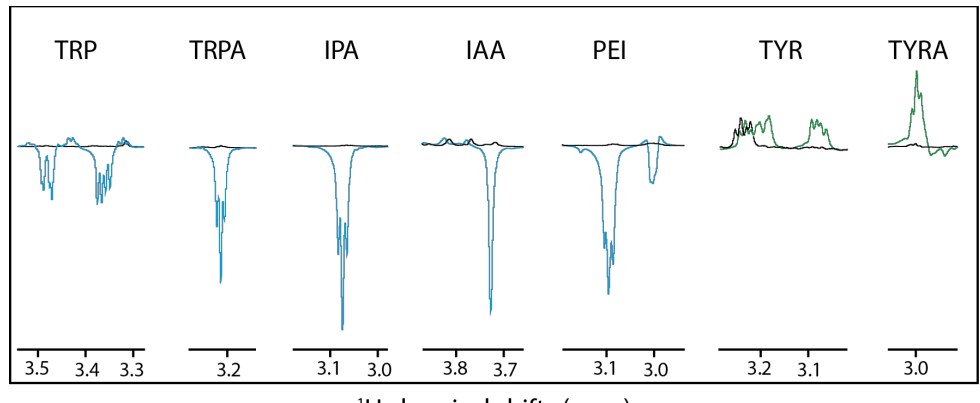

$^1$H chemical shifts (ppm)

**Figure 3: H$_\beta$ anomalous signals for the different molecules. Black lines are the reference spectra and blue lines are the irradiated spectra in the presence of fluorescein, and the green lines are the irradiated spectra in the presence of AT12. Scales between the different molecules are not respected for clarity purposes.**

The positive influence of hydrophobicity on the SNE may be explained by two distinct mechanisms that also may work in concert. First, the aromatic nature of the dye-molecule interaction is favored for more hydrophobic molecules. Second, the water shell surrounding the molecule is perturbed by the different hydrophobicity of the side-chains as it can be observed from the H$_\beta$ dynamics (Figure 3). Hence, the π-π stacking between the excited dye and the molecule, and therefore the orbital overlap, could be altered in a positive manner by increasing the hydrophobicity of the molecule. In summary, the hydrophobicity variation upon side-chain modification appears to have a qualitative impact on the CW-photo-CIDNP SNE unlike the charge and dynamic variation. With other words, the observed trends suggest a positive impact on the SNE for higher hydrophobicity of the molecules sharing a common aromatic moiety. While noting these findings, it must be stated (as above) that the exact nature of the polarization can only be determined by TR-photo-CIDNP. However, the presented empirical approach is regarded informative for CW-photo-CIDNP applications as the positive aspects of CW-photo-CIDNP with several

seconds of light irradiation in terms of signal to noise and easy and cheap set up is apparent. The importance on the irradiation time is indicated in Supplementary Figure 1 yielding for both the compounds tested (i.e. HOPI and TRP) an enhanced signal by a factor of 1.5 to more than 2 between one second and four second irradiation time.

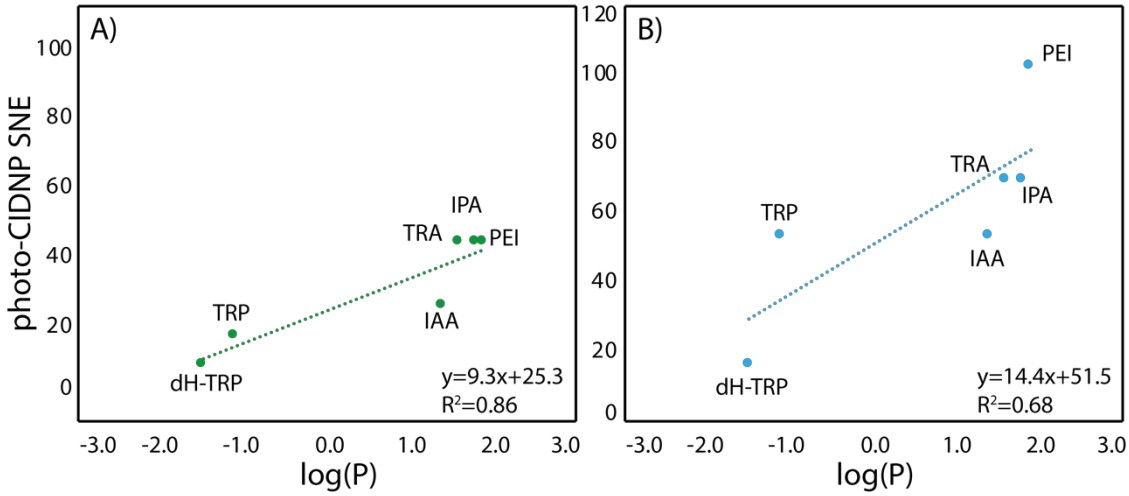

**Figure 4: Correlation between the molecule hydrophobicity and the photo-CIDNP performances for the tryptophan derivatives. A) In the case where CW-photo-CIDNP is monitored by AT12. B) In the case where CW-photo-CIDNP is monitored by fluorescein. Log(P) is the logarithm of the partition coefficient, P, between octanol and water and were calculated with DataWarrior®. A statistical analysis of the trendlines using Pearson's R coefficient and the Student t-test for the hypothesis shows that for AT12 (A) a t of 4.87 and a p-value of 0.008 and for fluorescein (B) a t of 2.6 and a p-value of 0.060 is obtained. Hence, in both cases, the hypothesis H0 (absence of correlation) is rejected, and the alternative hypothesis (correlation is non-null) is retained.**

In addition, the pH dependency of the photo-CIDNP SNE where recorded for the different tryptophan analogues, in a pH range between 5 and 9 within 2 units proximity of physiological conditions. While the pKa of the indole is typically around 16, the pKa of the indolyl group has been observed to be rather in the range 7-8 (Stob and Kaptein, 1989; Hore and Broadwurst, 1993). The latter pKa is known to have a significant influence on the photo-CIDNP hyperpolarization performances. The pH-dependent photo-CIDNP performances of the five tryptophan analogues (TRP, TRA, PEI, IPA, and IAA) for both dyes show overall similar behaviors but group in presence of fluorescence into TRP and TRA with a maximum already at pH 7 and IPA, PEI, and IAA with a maximum at pH 8 while the relative enhancement of IAA at pH 7 is significantly lower when compared with the other compounds (Figure S2B). In presence of AT12 the differences are less manifested with a maximum enhancement reached for pH > 9 (Figure S2B). Similar results have been observed by Stob and Kaptein, for tryptophan and N-acetyl tryptophan. Importantly, the log(P) dependencies were still observed at the optimal pH hereabove mentioned as the plots in Figure S3 show similar correlation as observed in Figure 4, at the difference that the IPA and IAA show the best enhancements for the photo-CIDNP spectra monitored with AT12 and fluorescein. While the common feature between these two compounds is the carboxylic acid ending the sidechain, the ionic interaction with the dye cannot explain simply these better performances, as the dye charges are significantly different (Figure 1). Moreover, the significantly better SNE for PEI as compared to TRA and TRP is in favor of a positive influence of the hydrophobicity. The extension of this hypothesis to a

broader set of molecules may be beneficial to confirm the positive impact of the carboxylic acid in comparison to other negatively charged molecules, and the impact of hydrophobicity on the photo-CIDNP performances.

## Conclusion

Photo-CIDNP appears to be an interesting approach for NMR signal enhancement through polarisation with the potential to rescue the low sensitivity usually inherent with NMR. The small library of CW-photo-CIDNP-active compounds presented indicates the existence of many photo-CIDNP active molecules in the known chemical space, mainly described in two prior studies (Stob and Kaptein, 1989; Hore and Broadwurst, 1993). The initial qualitative physico-chemical analysis of the 10 compounds studied indicate that side chains of the different molecules play a key role in the CW-photo-CIDNP SNE, which could not be predicted considering solely the ionic interaction monitoring the dye quenching or the radical pair stabilization. Rather the hydrophobicity of the molecules revealed to have an influence in SNE because the polarization performances improved gradually within the two class of derivatives with hydrophobicity. Importantly, the pKa of the indolyl radical is a critical parameter for the optimization of the polarization effect of photo-CIDNP. For indole groups, the optimal pH is above 8.0 for fluorescein and 9.0 for AT12, the measurement at higher pH improve in particular the polarization of the analogues carrying a carboxylic acid, which were found to be performing best in basic conditions. The importance of the hydrophobicity as one signature opens the way for simple chemical space exploration, since it can be simply assessed by its log(P). This allows for the possibility of empirically screening the chemical space for potential highly CW-photo-CIDNP-active small molecules. From this initial study, emerges the potential of making CW-photo-CIDNP a much broader method for signal enhancement in the biomedical NMR field, and the request for the exploration of the non-aromatic chemical-space within photo-CIDNP-active molecules.

## Material and Methods

The NMR measurements were performed at 298K either on a Bruker Avance III 600 MHz spectrometer equipped with a cryoprobe. The irradiation of AT12 samples was performed with a Coherent Verdi V10 diode pumped solid state laser emitting at a wavelength of 532 nm. The laser used for the fluorescein samples was a Thorlabs L450P1600MM, a diode laser emitting at 450 nm. The laser light was coupled (using appropriate coupling optics) into an optical fiber (Thorlabs, FG950UEC) of length 10 m and a diameter of 0.95 mm. The end of the fiber was inserted into the sample solution in a 3 mm NMR tube to a depth of about 5 mm above the NMR coil region.

Tyrosine, tyramine, tryptophan, tryptamine, IPA, IAA were purchased from Sigma, dH-TRP was purchased from Akos Pharma, and PEI was purchased from ChemSpace LLC. HOPI was synthesized in-house according to the previously published protocol (Torres et al., 2021). were prepared as stock solutions of 0.2 mg/ml and 0.18 mg/ml, respectively, in a 0.1 M

sodium/potassium phosphate buffer (pH 7.1) with 5% $D_2O$. The stock solution of AttoThio 12 (AT12) was 1 mg/ml in $H_2O$.

To prevent dye quenching the enzyme cocktail Glucose oxidase (Go, 120 kDa), catalase (Cat, 240 kDa) and D-Glucose (G,

180 Da) was used at a concentration of 14 nM for each enzyme and 2.5 mM of Glucose, as described elsewhere (Okuno and

Cavagnero, 2016;Lee and Cavagnero, 2013). The stock solutions were 0.25 μM for Go and 0.16 μM for Cat, respectively. The

glucose stock solution was 500 mM in $D_2O$ with 0.02% NaN3. All the samples were prepared in a 100 mM $KPO_4$ buffer at pH

= 7.1 with either 20 μM AT12 or 25 μM fluorescein, and 100 μM target molecule. The pH titrations where performed in the

same buffer, with the adjusted pH (5.0, 6.0, 7.0, 8.0, 9.0), and the oxygen scavenging was performed using a cycle of vacuum

and nitrogen atmosphere flush, for 30 min.

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
