# Peer review of "Exploration of the close chemical space of tryptophan and tyrosine reveals importance of hydrophobicity in CW-photo-CIDNP performances"

_Magnetic Resonance, 2021_

## Referee Comment (RC1)

**Exploration of the close chemical space of tryptophan and tyrosine reveals importance of hydrophobicity in photo-CIDNP performances**

**Torres, Renn, Riek**

I see no reason why this manuscript should not be published in its present form. The authors might like to make the following minor corrections.

Line 7:  conventionally, CIDNP is Chemically Induced … not Chemical Induced …

Line 13: replace extend by extent

Line 31: Kaptein et al. (1978) describes the use of photo-CIDNP to study protein structure, not protein folding. Articles on protein folding include:

> *J. Amer. Chem. Soc.* **119** (1997) 5049-5050
>
> *J. Amer. Chem. Soc.* **125** (2003) 12484-12492
>
> *Methods* **34** (2004) 75-87

A few abbreviations are used without explanation: HOPI, CW, ET

TCBP is first used as an abbreviation in line 48 but is not defined until line 125.

FLUO is defined as an abbreviation in line 125 but is never used.

Lines 54 and 69: constant wave should be continuous wave

Line 72: replace expends by extends

The text refers to parts A, B and F of Figure 1. The Figure itself has no such labels.

Line 98: replace spin-orbital by spin-orbit

Table 1:  the quantity Log(P) should be defined in the caption.

Figure 1: the (coloured) polarized spectra would be easier to see without the dark spectra which are so weak as to be almost completely flat.

Line 188: replace favorized by favoured or favoured

---

## Author Response (AR1)

**Laboratory for Physical Chemistry**

Eidgenössische Technische Hochschule Zürich
Swiss Federal Institute of Technology Zurich

ETH Zürich
HCI F 225
Wolfgang-Pauli-Str. 10
8093 Zurich
Switzerland

**Roland Riek, PhD**
Professor for Physical Chemistry
+41 44 632 6 139
+41 44 633 14 48
roland.riek@phys.chem.ethz.ch

Prof. Dr. Rolf Boelens
Editor

Magnetic Resonance

Zurich, February 2021

**Re: Submission of the revised version of the manuscript entitled "Exploration of the close chemical space of tryptophan and tyrosine reveals importance of hydrophobicity in CW-photo-CIDNP performances" by Torres et al.**

Dear Rolf

Please find therein the revised version of the manuscript entitled "Exploration of the close chemical space of tryptophan and tyrosine reveals importance of hydrophobicity in CW-photo-CIDNP performances" by Torres et al. for publication in your esteemed journal within the Festschrift to the birthday of Rob Kaptein. The manuscript has been evaluated by two reviewers. While reviewer (Dr. P. Hore) supports the manuscript strongly with minor corrections, the second reviewer rejects the manuscript stating that the mechanism of photo-CIDNP can only be elucidated if it is recorded time-resolved. In principle we agree with the reviewer's point that the determination of the mechanism of photo-CIDNP requests time-resolved experiments, but this was not the focus of our manuscript. The focus of our manuscript is an empirical chemical approach in building up a CW-photo-CIDNP active compound library. This manuscript is the initiation of such a library (with 10 compounds) and meanwhile we have obtained more than 100 CIDNP-active compounds with the aim to get several thousands. In our view for this a fast empirical approach within CW-photo-CIDNP as documented in the manuscript is necessary and will possibly allow for a machine-learning approach to elucidate CIDNP-active compounds.

Reading the criticism of the reviewer 2 the manuscript has been significantly revised (in addition to reviewer 1's comments). This includes (in short, and in details in the answering part of the reviewers)

(i)     Stating that the focus was an empirical approach within CW-photo-CIDNP and using everywhere CW-photo-CIDNP as the method where the findings are applicable.
(ii)    A statistical analysis of the findings in Figure 4.
(iii)   New data (presented in Suppl. Figure 1) showing the time-irradiation dependent signal enhancement obtained highlighting the importance of the use of continuous wave irradiation for highest signal to noise.

I sincerely hope that the revision is well suited for a publication as it is together with another manuscript (just accepted) our initial work on CIDNP and we plan to expand on it. We know that our point of view is orthogonal to the community's point of view (as one can see by the strong reaction of the reviewer), but of course we also want to elaborate on a different avenue than the others and time will show whether we will succeed – this is one of the excitement of science, isn't it?

Sincerely,

Point by point response to the requests and suggestions of the reviewers

We would like to thank the reviewers for the careful reading and interesting comments. While reviewer 1 is overall very positive and his minor requests were (with one exception) incorporated into the revised version of the manuscript, reviewer 2 is very negative. The statement of reviewer 2 in short is that the use of constant-wave (CW) photo-CIDNP is irrelevant while we are absolutely convinced that the approach we have is relevant in the purpose of our ultimate goal in bringing CW photo CIDNP into the field of biomedical research. The different views we attribute to the point of perspective. From a physical point of view (which is the one of reviewer 2) we fully agree that the elucidation of the mechanism of photo-CIDNP which is a highly complex interplay between two molecules can only be elucidated with time resolved photo-CIDNP (which we recently have done in collaboration with Alexandra Yurkovskaya for the specific molecule HOPI). However, when exploring the chemical space towards the elucidation of (many) highly active photo-CIDNP active compounds at concentration interesting for biomedical research requiring CW-photo-CIDNP, a screening approach is an alternative allowing for the establishment of a qualitative correlation between properties and polarization observed (our approach).
Please allow us to provide more detailed explanations for our perspective. We sincerely hope this will change your mind on the global significance of this work.

Reviewer 1:

*Line 7: conventionally, CIDNP is Chemically Induced ... not Chemical Induced ...*

*Line 13: replace extend by extent*

*Line 31: Kaptein et al. (1978) describes the use of photo-CIDNP to study protein structure, not protein folding. Articles on protein folding include: J. Amer. Chem. Soc.119 (1997) 5049-5050 J. Amer. Chem. Soc.125 (2003) 12484-12492 Methods34 (2004) 75-87*

*A few abbreviations are used without explanation: HOPI, CW, ET TCBP is first used as an abbreviation in line 48 but is not defined until line 125.*

*FLUO is defined as an abbreviation in line 125 but is never used.*

*Lines 54 and 69: constant wave should be continuous wave*

*Line 72: replace expends by extends The text refers to parts A, B and F of Figure 1. The Figure itself has no such labels.*

*Line 98: replace spin-orbital by spin-orbit*

*Table 1: the quantity Log(P) should be defined in the caption.*

*Figure 1: the (coloured) polarized spectra would be easier to see without the dark spectra which are so weak as to be almost completely flat.*

*Line 188: replace favorized by favoured or favoured*

Answer: All suggestions were incorporated into the revised version of the manuscript with the exception of the suggestion on Figure 1 to remove the dark spectra. We think it is important to see the signal enhancement between the dark spectrum (practically flat) and the enhanced spectrum. We understand that there is little to see, but this is attributed to the relatively many data presented and we like to show data. We hope the reviewer is o.k. with this view, otherwise we would follow his suggestion.

Reviewer 2:    *"his qualitative study of hydrophobicity could only be relevant, if the authors had applied time resolved detection and analyzed the CIDNP signals in the products of geminate recombination, often called geminate CIDNP. Under continuous illumination, however, the resulting CIDNP effects are formed as a result of a complex interplay of spin and molecular dynamics of free radicals, but it is wrong to predict CIDNP intensities based alone on the properties of the reacting diamagnetic molecules as it was done in the manuscript. As is known, some properties of free radicals can be very different from the properties of molecules in the diamagnetic state. For example, the pKa values of radicals (tryptophan and tyrosine) and the same molecules in the diamagnetic state differ by several units. Spin is the magnetic moment of nuclei and electrons; therefore, the "key players" in the formation of geminate CIDNP are the magnetic properties of radicals, as shown by R. Kaptein, namely the hyperfine coupling (HFC) constants, the difference in the g-factor of radicals in the spin-correlated pair, and the magnetic field strength. Polarization in an F-pair, as is also well known, depends on the competition between the rate of paramagnetic nuclear relaxation and the bimolecular rate of radical recombination. They can be different too for the set of chosen compounds. In addition, paramagnetic nuclear relaxation in radicals is determined by the anisotropy of the HFC. None of these parameters is mentioned in the manuscript for the ten compounds studied. I'd like to stress, that Kaptein's work in the field of CIDNP goes much deeper than the simple sign rules"*

Answer: The reviewer points out with clarity the theoretical frame of photo-CIDNP. He demonstrates the difficulty to correlate the intensity of the spin sorting happening during to geminate polarization and the final intensity when light is irradiated for several seconds with as part free radical encountering (F-pair). Therefore, the reviewer rejects the interpretation of CW-photo-CIDNP results as they may not correspond to what would be observed from TR-photo-CIDNP experiments. However, the authors have no such intention of performing TR-photo-CIDNP experiments to explore fine radical-pair mechanism. On contrary, the goal of the authors is to explore the possibility of implementing CW-photo-CIDNP as a simple method to obtain hyperpolarization in solution state biomedical NMR. The reason for our choice in using solely CW-photo-CIDNP are the following:

1) We want to promote photo-CIDNP as a readily implementable technique for solution state biomedical NMR polarization. CW-lasers prices are relatively attractive (2000 USD) and extremely simple of use. Because of this, CW-photo-CIDNP impact could be important for the biomedical NMR community contrasting the very expensive DNP approaches.

2) The concentrations used are significantly lower than what is typically used in TR-photo-CIDNP. As an example, while we use 0.1 mM of molecule and 0.02 mM of photosensitizer, Saprygina et al. Use 1.1 to 40 mM of molecule and 2 mM of photosensitizer (Figure 4 of *Saprygina et al. J. Phys. Chem. A 2014, 118, 339-349*). In biomedical research the use of molecule concentration in the range of a few microM is essential and was achieved by other studies such as Okuno et al. J Phys Chem B, 2016, 715-723, who demonstrated the use of CW-photo-CIDNP to detect low micro-molar concentration of tryptophan with fluorescein. For this reason, we perform CW-photo-CIDNP, to reach the maximal polarization achievable to measure the molecules and not to understand the mechanism of polarization. It is about applying photo-CIDNP. We are happy to share the polarization build-ups with increasing irradiation time to support our statement in a revised version of the manuscript.

3) The scope of the study is to answer the question: what is the impact of chemical modification on the polarization performance of CW-photo-CIDNP? We provide here a view on the effect of side chain modification on the performances. We provide here a result which is the trend of the performance according to hydrophobicity in CW-photo-CIDNP and with this approach we were successful in finding the compound 3-(2-(piperazin)ethyl)-indole to be polarizable by a factor of 100 at 600 MHz on 1H just by screening.

4) As there is no simple correlation between TR-photo-CIDNP and CW-photo-CIDNP results, and we are interested in the performance of CW-photo-CIDNP. It would be irrelevant to interpret results from TR-photo-CIDNP to predict what would happen in the CW regime. For this reason, we opted for an empirical

approach consisting in exploring the chemical space in the conditions of the final application which is CW-photo-CIDNP.

5) The presented manuscript is the starting point to build up a large library of CW-photo-CIDNP active compounds and for large we mean at least a 1000 compounds to be applicable in biomedical research. Meanwhile with the use of the findings within the manuscript we have now already more than 1000 compounds identified to be tested.

We regret that we may have not make this perspective clear enough in our manuscript. In the revised version of the manuscript, we tried to highlight this approach much clearer by

- stating everywhere CW-photo-CIDNP as the method of choice also including the title and the abstract.

- Stating that the mechanism of action can only be elucidated with time-resolved photo CIDNP in the introduction as well as at the end of the discussion

  o "While the exact mechanism of polarization can be only evaluated by TR-photo-CIDNP, in the context of biomedical NMR application aiming for the highest polarization at low micromolar molecule concentration, CW-photo-CIDNP appears to be the method of choice suggesting the exploration of a CW-photo-CIDNP-based empirical approach indicated."

  o "While noting these findings, it must be stated (as above) that the exact nature of the polarization can only be determined by TR-photo-CIDNP"

*Reviewer 2: "The CIDNP method in its continuous mode should be applied with great care when used for any quantitative analysis. The concentration of radical pairs in the reaction of a triplet dye with a quencher essentially depends on the quenching rate constant. In the cited work of Saprygina et al. J. Phys. Chem. A 2014, 118, 339-349, the quenching rate constants were obtained from optical studies, the conditions for the time resolved experiment were carefully adjusted, and only geminate CIDNP was considered for quantitative analysis of the pH dependence. This was not done in the paper under review."*

Answer: This study, conducted under TR-photo-CIDNP, is focusing on the quenching rates in conditions relatively far than the conditions we have been using during this study. The concentrations are significantly higher (see point 2) and the type of dye is different. Indeed, the TCBP which is used, possessed 4 negatively charged carboxylates, the dyes that we have been using are significantly less charged.

*Reviewer 2: "There is a well-documented study where the terms "hydrophobicity" and "hydrophobic collapse" were shown to be misleading for explanation of the absence of CIDNP signals of tryptophan residues in the unfolded HEWL protein under cw-illumination. This work is cited as reference 8 in the reviewed manuscript. In ref 8, the time-resolved CIDNP detection revealed CIDNP tryptophan signals of similar strength at the geminate stage for the unfolded and native state of HEWL. It means, that hydrophobicity was not a relevant parameter for the description of CIDNP in the native and unfolded state of HEWL. Instead, the reaction of intramolecular electron transfer from tyrosine to the tryptophan radical on a microsecond time scale in the unfolded protein was found to be the main cause of a decrease in the Trp signal and an increase in the tyrosine signal."*

Answer: This interesting case study demonstrates the presence of intramolecular electron transfer and its effect on photo-CIDNP anomalous lines intensities. However, the phenomenon described here does not apply to our study, as we do not have intramolecular electron transfer. Moreover, we took care of comparing only molecules sharing the same aromatic system in order to minimize the difference in the magnetic parameters. This idea is already present in the paper from Saprygina et al., when the effect of charge is compared by modifying the side chain.

*Reviewer 2: "A direct comparison of the CIDNP data obtained under continuous illumination without measuring the quenching rate constant is inappropriate, since different concentrations of the quencher and different rate constants lead to different concentrations of the formed pair of geminate radicals."*

Answer: As stated before, this is why we focus our work on chemical exploration and we take an empirical approach. A bottom-up approach (from ns to second irradiation) will find potentially many ambushes in the context of chemical space exploration due to the complexity of the mechanism. Towards the application of CW-photo-CIDNP in biomedical research we envision low concentrations of molecules and always the same concentrations for evaluating a KD are required.

It is further noted, that the quenching rates are not measured in the present manuscript. The importance of CW-photo-CIDNP in potential biomedical applications where the signal is the limiting factor is further indicated in Suppl. Figure 1 in which the signal enhancement in respect of the irradiation time measured is shown.

*Reviewer 2: "I highly recommend reading the following articles by Robert Kaptein: Kaptein, R.; Den Hollander, J. A. Chemically induced dynamic nuclear polarization. X. On the magnetic field dependence. J. Am. Chem. Soc. 1972, 94 (18), 6269-80. Kaptein R. Chemically induced dynamic nuclear polarization. VIII. Spin dynamics and diffusion of radical pairs. J. Am. Chem. Soc. 1972, 94 (18), 6251-62. Stob, S.; Kaptein R. Photo-CIDNP of the amino acids. Photochem. Photobiol. 1989, 49 (5), 565-577."*

Answer: We would like to thank the reviewer for the relevant suggested reading. Field dependency, with the two different dyes would be interesting to demonstrate the equivalence of the magnetic parameters, however we do not have a spectrometer adapted to this kind of study, but we are in contact with collaborators in this context for another study. Here, we initiate the establishment of a CW-photo-CIDNP library, which we build up.

*Reviewer 2: "The over-interpretation of the small data set as presented here is misleading. Also, the historical overview of Kaptein's contribution to CIDNP theory is rather superficial, because it does not go deeper than just application his simple rule for the polarization sign.*

*With my regret, I recommend to reject the manuscript in its present form."*

Answer: As of the data set, we did a statistical test to demonstrate the significance of the trend line. A statistical analysis of the trendlines using Pearson's R coefficient and the Student t-test for the hypothesis shows that for AT12 (A) a t of 4.87 and a p-value of 0.008 and for fluorescein (B) a t of 2.6 and a p-value of 0.060 is obtained (as written in the caption of Figure 4).

We are sorry to see that the illustration of the Kaptein's rules were not appreciated. To our knowledge, the HOPI and the dH-TRP are the only biological molecules for which the polarization sign alternates when the dye is switched.

We hope that revised version of the manuscript highlighting the presence of an empirical approach, stating that the mechanism of action can only be revealed by time-sensitive CIDNP and that the applicability of the empirical approach is only valid within the CW-photo-CIDNP along with the statistical analysis, the measurement of CW-dependent polarization in supplementary figure 1 to highlight the importance of CW-photo-CIDNP in potential biomedical applications where signal is the limiting factor will change his opinion on our work.

---

## Author Response (AR2)

Eidgenössische Technische Hochschule Zürich
Swiss Federal Institute of Technology Zurich

**Laboratory for Physical Chemistry**

ETH Zürich
HCI F 225
Wolfgang-Pauli-Str. 10
8093 Zurich
Switzerland

**Roland Riek, PhD**
Professor for Physical Chemistry
+41 44 632 6 139
+41 44 633 14 48
roland.riek@phys.chem.ethz.ch

Prof. Dr. Jörg Matysik
Editor

Magnetic Resonance

Zurich, February 2021

**Re: Submission of the second revised version of the manuscript entitled "Exploration of the close chemical space of tryptophan and tyrosine reveals importance of hydrophobicity in CW-photo-CIDNP performances" by Torres et al.**

Dear Jörg

Please find therein the second revised version of the manuscript entitled "Exploration of the close chemical space of tryptophan and tyrosine reveals importance of hydrophobicity in CW-photo-CIDNP performances" by Torres et al. for publication in your esteemed journal within the Festschrift to the birthday of Rob Kaptein. The revised manuscript has been evaluated by two additional reviewers (in addition to the two initial reviews). While reviewer 3 had only minor corrections reviewer 4 placed the attention to potential different pKa effects of the various compounds with the potential to interfere with the findings on the hydrophobicity SNE correlation found at pH 7. While this suggestion is well funded and also observed in a pH titration study performed (Figure S2) the correlation between SNE and hydrophobicity is still observed well above the pH sensitive region of the SNE (Figure S3). Please find more detailed answers to the questions of the reviewer below.

I sincerely hope that the second revision is now well suited for publication.

Sincerely,

Point by point response to the requests and suggestions of the reviewers

Reviewer #3
Reviewer #3: The term CV should be used in this common notation as continuous wave not as constant wave as on page 3, line 68, for example.

Answer: We would like to thank the reviewer for careful reading. Corrections are done as suggested.

Reviewer #4
Reviewer: The manuscript describes the exploration of chemical space of two aromatic amino acids in CW-photo-CIDNP. Whereas TR-photo-CIDNP would be mechanistically more informative, the approach using CW-photo-CIDNP has clear sensitivity advantages. With their approach the authors could demonstrate the existence of several compounds showing strong photo-CIDNP effects.

In cyclic reactions, as studied here, cancellation of polarization of recombination and escape products (chemically the same in a cyclic reaction) will occur. The observed CIDNP depends critically on the various kinetic rates and nuclear spin relaxation times. Thus not only for understanding the mechanism, but also for optimal CW-CIDNP intensity and identifying new compounds exploration of experimental conditions can play a significant role. The authors could comment on this.

Answer: We agree with this statement of the reviewer that the exploration of experimental conditions are important. In this context, we extended in the revised version the experimental conditions to a pH range plus/minus 2 units around the physiological pH of 7 (Figure S2 and S3). Our interest in exploring CIDNP in drug related research constraints the conditions however significantly (including in addition to a pH around 7-8 also low compound concentration).

Reviewer: Recommendable overviews for this study would be Hore and Broadhurst (1993), Photo-CIDNP of Biopolymers, Prog. NMR Spectrosc. 25, 345-402 and Kuhn (2013), Photo-CIDNP NMR Spectroscopy of Amino Acids and Proteins. In: Kuhn (eds) Hyperpolarization Methods in NMR Spectroscopy. Topics in Current Chemistry Vol 338, Springer Berlin Heidelberg, pp. 229-300, https://doi.org/10.1007/128_2013_427
A related study showing examples CW-photo-CIDNP of tyrosine and tryptophan derivatives would be Stob and Kaptein (1989), Photo-CIDNP of the amino-acids, Photochem. Photobiol. 49(5):565-577, https://doi.org/10.1111/j.1751-1097.1989.tb08425.x. Also the review by Hore and Broadhurst summarizes several dyes and compounds for photo-CIDNP.

Answer: We would like to thank the reviewer for the references, which are now incorporated into the manuscript.

Reviewer: Scheme II and scheme I in the paper by Stob and Kaptein may provide a simple framework to interpret the results in this manuscript as well. That paper also shows a pH and concentration dependency of the CW CIDNP intensity. Fig. 15 in that paper may be a warning, since small [Trp] concentrations can have a steep CIDNP intensity effect, idem Fig. 14 showing strong pH dependency at pH 6. Also the review by Hore and Broadhurst shows such pH dependency for Tyr and Trp (Fig.6, and explained on p.363). The referred paper by Okuno and Cavagnero (2016, J Phys Chem B) also shows possibilities for simulating CW-photo-CIDNP and parameters for that.

The pH dependency of CW-photo-CIDNP intensities could also deviate substantially from the pKa's of compounds. Thus characterizing CW-photo-CIDNP at a few concentrations and pH values may be recommendable for this type of screening. Whether concentration and pH could indeed play a role, the authors could partially check this already using the equations given by Stob and Kaptein.

Answer: To elaborate on the pH effect of the CW-photo-CIDNP we extended the TRP analog studies over a pH range between 5-9 with both dies AT12 and fluorescein (Figure S2) knowing however that our interest is the physiological pH range 7-8. As expected by the reviewer and now discussed in the manuscript a strong pH effect is observed. In this study, we had however to change the oxygen scavenger system as the enzyme cocktail is limited to physiological conditions yielding an overall significant drop of the signal enhancement. As stated above by the reviewer, CW-photo-CIDNP depends on the conditions -).

Reviewer: Though not really observable when perfect 90° detection pulses were used (better detected with 45° pulses), the spectral difference between Tyr and TyrA may also point to a difference in a multiplet effect, for which there is another Kaptein' CIDNP rule, and which would be independend of a diference in g-values in the radical pair and may therefore still be present in case of weak net effects.

Answer: Following the hypothesis of the reviewer we measured the spectra with 45 degree pulse (Figure S4 and Figure S5 below) not finding indication that would support this hypothesis. This experiments were not taken up in the manuscript.

[Figure]

Figure S1: Photo-CIDNP spectra of tyrosine (Tyr) and tyramine (TyrA), with 45° detection pulses. Zoom on the aromatics. The samples were concentrated at 100 μM of Tyr/TyrA and 25 μM of AT12.

[Figure]

Figure S2: Photo-CIDNP spectra of tyrosine (Tyr) and tyramine (TyrA), with 45° detection pulses. Zoom on the aliphatics. The samples were concentrated at 100 μM of Tyr/TyrA and 25 μM of AT12**.**

Reviewer: In Table 1, I would use / add the same short names and order as in Fig. 1 (then short names are also defined, and not partially as now in Fig. 2). Please check SNE's in Fig .1 and Table 1: e.g. AT12, SNE for TyrA is -60 (Table 1) and -63 (Fig. 1), whereas with the other molecules numbers appear to be the same. Conditions for the experiment are also the same.

Answer: We would like to thank the reviewer for his careful reading. We followed his suggestions.

Reviewer: The authors may also consider at several instances pKa changes also source for the CIDNP intensity differences:
- Could pKa differences also be explanation for CIDNP intensity changes between HOPI / dH-Trp, IPA / IAA, etc?

Answer: Indeed, based on the new measurements at various pHs, pKa differences are a partial explanation to the differences in the photo-CIDNP intensities, especially in the case of the IAA and IPA. However, the log(P) contribution tendency was still present at different pH (Figure S3, Figure 4).

Reviewer: I. 168: interestingly, such pKa changes had been discussed by Stob and Kaptein (1989) for Trp and N-acetyl-Trp. Could such pKa play a role here as well?

Answer: The differences are between TRP and indole/N-methyl-TRP. The N-acetyl and TRP show quite a small difference especially if one considers the accuracy of the measurement exhibited in the figure 5 of the Stob and Kaptein paper.

Reviewer: l.184/5: are these the estimated pKa's for fluorescein that of its radical? Idem for the other (notably 'tryptophan') pKa's on p.8.

Answer: The pKa here mentioned are the pKa of the ground state molecules and now stated as such in the manuscript. We considered these ones in this part as we discuss the quenching rates which are before the formation of the radical pair.

Reviewer: l. 205: please note that the precise pKa value and thus changes therein could also play a significant role on the CW-CIDNP effects studied here as well. That would much more subtile than overall charge, and could also be explanation for failure of just using charge alone.

Answer: This suggestion was put in practice by measuring the pH titration, and the log(P) dependencies at different pH (Figure S2 and S3). Please see above answers for more details. It was interesting to see that the carboxylic containing analogues are actually the best polarized.

Reviewer: l. 210: the rates of sidechain dynamics may be much slower than the various rates in CIDNP (reactions, ET, protonation). Dynamics would then rather be: presenting more or less active or conformations, or conformations with different pKa's.

Answer: This is an interesting point. The pKa of interest here is the indolyl pKa. The measurement at high pH (i.e. pH 9) where the protonation of the indolyl was everywhere the same should give insights into this hypothesis. As found in Figure S3 the effect under investigation was still observed at high pH challenging the hypothesis. The sidechains dynamics may be potentially involved in the rearrangement dynamics of the water shell around the molecule. This could have an effect on the radical pair formation. This is however pure hypothesis but would fit with the Marcus theory.

Reviewer: l. 222/ Fig.4: the observed correlation of CIDNP SNE and hydrophobicity is interesting. But possibly pKa effects present also a good explanation for observed effetcts. The authors could discuss this as alternative to hydrophobicity.

Answer: As mentioned above already several time this important suggestion was put into practice by measuring the pH titration (Figure S2), and the log(P) dependencies at different pH (Figure S3) and by a new paragraph reading:

"In addition, the pH dependency of the photo-CIDNP SNE where recorded for the different tryptophan analogues, in a pH range between 5 and 9 within 2 units proximity of physiological conditions. While the pKa of the indole is typically around 16, the pKa of the indolyl group has been observed to be rather in the range 7-8 [Kaptein Stob ref]. The latter pKa is known to have a significant influence on the photo-CIDNP hyperpolarization performances. The pH-dependent photo-CIDNP performances of the five tryptophan analogues (TRP, TRA, PEI, IPA, and IAA) for both dyes show overall similar behaviors but group in presence of fluorescence into TRP and TRA with a maximum already at pH 7 and IPA, PEI, and IAA with a maximum at pH 8 while the relative enhancement of IAA at pH 7 is significantly lower when compared with the other compounds (Figure S2B). In presence of AT12 the differences are less manifested with a maximum enhancement reached for pH > 9 (Figure S2B). Similar results have been observed by Stob and Kaptein, for tryptophan and N-acetyl tryptophan. Importantly, the log(P) dependencies were still observed at the optimal pH hereabove mentioned as the plots in Figure S3 show similar correlation as observed in Figure 4, at the difference that the IPA and IAA show the best enhancements for the photo-CIDNP spectra monitored with AT12 and fluorescein. While the common

feature between these two compounds is the carboxylic acid ending the sidechain, the ionic interaction with the dye cannot explain simply these better performances, as the dye charges are significantly different (Figure 1). Moreover, the significantly better SNE for PEI as compared to TRA and TRP is in favor of a positive influence of the hydrophobicity. The extension of this hypothesis to a broader set of molecules may be beneficial to confirm the positive impact of the carboxylic acid in comparison to other negatively charged molecules, and the impact of hydrophobicity on the photo-CIDNP performances*.
"

Reviewer: 'Absolute' may still be hard, but would relative pKa predictions using DFT calculations of classes of 'tyrosine' and 'tryptophan' compounds be useful for interpreting the CIDNP intensities, as an additional approach to charge/log(P) or TR-photo-CIDNP?

Answer: We looked into potential calculations of the pKa with DFT and found that on top of its time-intensive computing nature the accuracy is not that great and that experiments can be done much faster. Since from an experimental group point of view the experiment is the gold standard, we decided to do the pH titration as shown in Figure S2.